# Relationship between Landform Development and Lake Water Recharge in the Badain Jaran Desert, China

**Da-Peng Yue [1], Jing-Bo Zhao [1,2,\*], Yan-Dong Ma [2,3] , Xiao-Gang Huang [1], Tian-Jie Shao [1], Xiao-Qing Luo [1] and Ai-Hua Ma [1]**

[1] School of Geography and Tourism, Shaanxi Normal University; Xi'an 710062, China; yuedp@snnu.edu.cn (D.-P.Y.); huangxg@sxnu.edu.cn (X.-G.H.); tjshao@snnu.edu.cn (T.-J.S.); luoxq0815@126.com (X.-Q.L.); maaihuaemail@163.com (A.-H.M.)

[2] State Key Laboratory of Loess and Quaternary Geology, Institute of Earth Environment, Chinese Academy of Sciences, Xi'an 710075, China; mayandongemail@163.com

[3] Key Laboratory of Subsurface Hydrology and Ecological Effects in Arid Region of the Ministry of Education, Chang'an University, Xi'an 710054, China

\* Correspondence: zhaojb@snnu.edu.cn

**Abstract:** Four distinctive but poorly documented landforms in the Badain Jaran megadunes were studied: arcuate steps, multi-stage fans, depressions formed by runoff erosion, and groundwater overflow zones around lakes. The development of these four landform types indicates the following: (1) The hydrological balance in the sand layers of the megadune areas is positive; (2) After evaporation and transpiration, precipitation is able to infiltrate the deep sand layers; (3) Precipitation is a source for the groundwater and for many of the lakes of the area. The groundwater recharge mechanism is characterized by intense precipitation events that provide a water source, high infiltration rate, shallow evaporation depth, and low water retention. These factors together enable the precipitation to be transformed into groundwater. The energy of gravity water and the high water film pressure of adsorbed water together provide the forces necessary for effective water recharge.

**Keywords:** desert landforms; groundwater overflow zone; gravity water; precipitated gypsum and calcite; recharge mechanism

---

## 1. Introduction

Numerous studies have been conducted on various aspects of the deserts in China. They include studies of geomorphic type, sand dune shape, and particle size characteristics [1–4]; and of wind direction and strength [4–8]. Granulometric studies of Chinese deserts have revealed the strength and direction of the prevailing winds, and indicated that the sediments are mainly fine sand. The Badain Jaran Desert has the coarsest grain-size composition in China, followed by the Tengger Desert [8,9]; whereas the Gurbantünggüt and Taklamakan deserts have the finest grain-size composition [1,5,9]. Studies of wind conditions have revealed that the Badain Jaran Desert has the highest average wind speed, followed by the Gurbantünggüt and Tengger deserts, while the Taklamakan Desert has the lowest [1,5]. There are substantial differences in wind energy conditions in different deserts. The predominant wind direction in China's deserts are westerly and northwesterly [6,7]; however, the Badain Jaran and Tengger deserts have a predominant southeasterly wind direction.

Deserts are among the most distinctive landscapes on earth. There are many types of sand dunes in deserts, with the largest mainly classified as compound. The megadunes in the Badain Jaran Desert are the highest on earth, although those in the Taklamakan Desert are also large. Lakes are often found within desert areas. The development of numerous small lakes in the Badain Jaran megadune area has

created a highly distinctive landscape. The lakes are primarily distributed at the bottom of the leeward slopes of the megadunes, which indicates that there may be a relationship between the development of the lakes and the adjacent megadune. The Badain Jaran Desert has received substantial research attention in recent years due to its megadunes and numerous lakes [6,10–13] and is the subject of the present study.

The Badain Jaran Desert has a unique landscape of sand dunes and numerous lakes, (>140) [4,14] and it has the highest megadunes on earth [4,7,8,14,15]. These prominent landscape features—megadunes and lakes—have attracted much research attention [3,11,12,16,17]. In particular, the origin of the lake water has been greatly debated in recent years. Despite the lack of direct evidence, several researchers have concluded that the lake water in the area is derived either from recent precipitation [18–20] or from precipitation during the Holocene [21,22]. However, others have claimed that the Qilian Mountains, which have modern glaciers, supply the lake water via an underground fault [13]. However, because the groundwater level in the Badain Jaran Desert is higher than that in the area between the Qilian Shan and the desert, it is impossible for the Qilian Mountains to be the source of the lake water [20].

The relatively complexity terrain and large elevation differences of the Badain Jaran Desert make it difficult to carry out field investigations. In addition, the central-northern part of the desert was poorly investigated, which is the reason why several of the landforms developed on the slopes of the megadunes were not described previously. During the past three years our research group has discovered multistage fan microtopography and secondary salinization formed by spring water seepage in the Badain Jaran Desert, and we have concluded that the lake water in the area is mainly supplied from precipitation [23]. However, we only discovered micro-landforms representing the late stage of spring water formation, and no evidence was found of microtopography representing the early stage of spring water formation. Most of the landforms discussed herein were not reported by Ma et al. [23]; in addition, Ma et al. [23] neither studied nor proposed a mechanism for groundwater recharge by precipitation. Here, we report the results of a detailed investigation of multiple water flow pathways in four megadunes (including Zhalate Lake East Megadune), which provides evidence for the recharge of lake water by precipitation. In the course of the investigation we discovered many previously undescribed landforms and our results provide a basis for understanding the role of precipitation in lake water recharge in the region, together with the mechanisms of groundwater recharge. Our findings are potentially applicable to other extremely arid areas.

## 2. Materials and Methods

The Badain Jaran Desert is in Alxa Youqi of the Inner Mongolia Autonomous region of China and is surrounded by the Ejina River to the west, Gurinai Lake to the north, Heli Hill of the Hexi Corridor to the south, and Zongnai Hill and Yabulai Mountain to the east. The sand dunes are widely distributed and are the highest on earth. The aeolian landforms are complex and include barchan dunes and barchan dune chains, transversal dunes, square dunes, scaled dunes, and pyramid dunes. In the southern part of the desert there are rolling megadunes and numerous lakes (Figure 1). The heights of the large dunes in the study region are mainly in the range of 200–500 m. The windward slopes of the combined barchan dunes contain superimposed secondary barchans. At the base of the dune slopes there are low plains with sand ripples that have gradients of 6°–9°. The lower parts of the megadunes are small barchan dunes, with height differences from several meters to tens of meters and gradients from 12° to 19°. The middle parts of the megadune slopes consist of chains of barchan and longitudinal dunes, with heights of several tens of meters and gradients of ~25°. The tops of the megadunes form peaks with heights of several 100 meters and with gradients of 30°–34°.

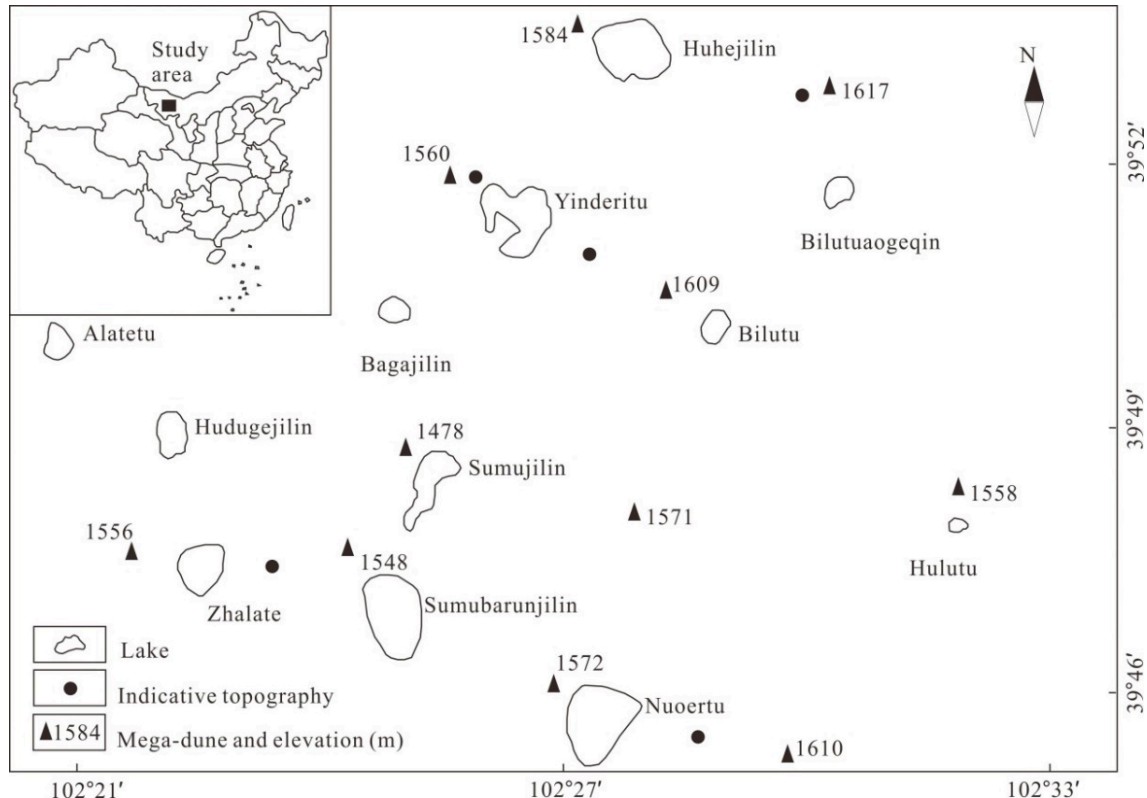

**Figure 1.** Location of the study area and sites of the studied multi-stage fans in the Badain Jaran Desert.

Several small lakes occur at the base of the leeward slopes of the megadunes, and their areas are mainly less than 1 km$^2$. Nuoertu Lake is the largest (1.45 km$^2$), with a depth of ~16 m. Controlled by the climate of high barometric pressure, the climate of the area is continental with cold and dry winters, extremely hot summers, and with spring and autumn seasons. The annual precipitation is 40–120 mm [20]. There are large differences in annual temperature and there is strong radiation. The region is influenced by northwesterly and westerly winds year-round, with average wind speeds of 3.0–4.5 m/s. During field investigations we discovered four previously undocumented landform types: arcuate steps, multi-stage fans, depressions formed by runoff erosion, and groundwater overflow zones. They occurred on the eastern megadune of Nuoertu Lake (39°46′ N, 102°29′ E), on the eastern megadune of Zhalate Lake (39°48′ N, 102°23′ E), on the eastern megadune of Huhejilin Lake (39°52′ N, 102°29′ E), on the western megadune of Yinderitu Lake (39°50′ N, 102°28′ E), and on the western megadune of Badain Jaran Lake (39°47′ N, 102°29′ E) (Figure 1). The eastern megadune of Nuoertu Lake is 1610 m above sea level (a.s.l.) and 420 m high; the eastern megadune of Zhalate Lake is 1548 m.a.s.l. and ~350 m high; the megadune of eastern Huhejilin Lake is 1617 m.a.s.l. and ~400 m high; and the megadune of western Huhejilin Lake is 1584 m.a.s.l. The western megadune of Badain Jaran Lake is 1478 m.a.s.l., and the megadune of western Yinderitu Lake is 1560 m.a.s.l. The four landform types can be observed in each month of the year and they are usually developed on the windward side of the dunes.

Two 1-m-deep profiles were dug in the middle of the 1610-m-high megadune east of Nuoertu Lake and in the lower part of the 1548-m-high megadune east of Zhalate Lake. At each site, 10 samples at a 10-cm interval were obtained for particle-size analysis. The samples encompass both the relatively coarse-grained surface later and the fine-grained deeper layer. In addition, eight samples of precipitated salts were collected from the eastern megadunes of Zhalate Lake and Huhejilin Lake. For comparison, on the windward slope of the eastern megadune of Nuoertu Lake, a depth profile of 52 samples was obtained for particle-size analysis from the coarse surficial sand layer (above 2 cm) and the underlying finer sand layer (below 10 cm).

In addition, four drill cores were taken from the windward slope of the megadune to the west of Huhejilin Lake for moisture content determinations and particle-size analysis. The cores were 4-m long and samples were collected every 10 cm (a total of 168 samples). The drilling holes of the upper section at 1245 m a.s.l. were labeled 'a' and 'b', and the lower holes at 1217 m.a.sl. were labeled 'c' and 'd'.

Particle-size analysis was carried out using a Mastersizer2000 laser particle-size analyzer (Malvern Company, Malvern, UK). The precipitated salt samples were analyzed by electron microscopy and EDX analysis. The water content was measured by drying and re-weighing. In addition, based on the 30-m resolution digital elevation model, the slope analysis module of ArcGIS 10.2 (developed by Environmental Systems Research Institute (ESRI) located in Redlands, California, US) was used to statistically analyze the slope of the megadune-lake area, and the area of slope with >20° was obtained. This area was then used to assess the water balance in the megadune-lake area. All of the analyses were conducted in the environmental science and physics laboratories of Shaanxi Normal University.

Water in the sand layer and soil layer is mainly divided into adsorbed water and gravity water [24–26]. Adsorbed water exists in the form of a water film and is thus termed film water. It is generally thought that if the water content of the sand layer is <3% then the water is adsorbed water, and if the water content is >3% then it is gravity water [27,28]. In this study, we assumed that values of <3.5% represented adsorbed water and values >3.5% represented gravity water.

## 3. Results

### *3.1. Arcuate Steps on the Slopes of Megadunes*

The arcuate stepson the slopes of megadunes have an approximate semi-arc shape with a bulge in the middle part and backward shrinkage at both sides. The length of the arc is 1–3 m and the height of the leading edge is 5–20 cm. Macroscopic observations indicated that the arcuate steps are composed of aeolian sand, with the surface commonly consisting of dark-colored silt (Figure 2a–c). Occasionally a moist surface could be observed in front of the arcuate landform (Figure 2b), and occasionally white salt precipitates were distributed on the surface (Figure 2c,d). These arcuate landforms are developed on the slopes of many megadunes, such as those to the east of Zhalate Lake and to the west of Yinderitu Lake. This micro-landform is evidently formed by water exudation.

### *3.2. Depressions Caused by Surface Runoff*

Small (~1-m diameter) depressions were observed (Figure 2e). They have a flat bottom and there is typically a moist surface layer with dark-colored silt accumulated at the base. The edge of the slope is gentle, with a gradient of 3°–10°. Rills resulting from runoff erosion are present on the edge of the depression, and there is a water outlet in the front of the bottom of the depression.

### *3.3. Multi-Stage fans Deposited by Spring Water*

We observed four multi-stage fans which were deposited by spring water. The areas were small (4–10 m$^2$) and they had a narrow top (≤0.5-m wide) and a wider front edge. The width was greatest in the middle and the front part was 2–4 m wide. The front edge formed an irregular and concave-convex circular arc, which was sometimes sinuous (Figure 2f–i). The deposits had clearly-defined bedding (Figure 2j). The trailing edge of the fans was high and the leading edge was low. The trailing edge was inclined forward slightly towards the leading edge with a gradient of 5°–10°, and the height difference between the leading and trailing edges was 0.5–2.0 m. The most distinctive characteristic of the fans was the numerous steps (up to 20) developing on the surface (Figure 2f–i); each step was very smooth, 5–30 cm wide and 3–10 cm high.

### *3.4. Secondary Formation of Carbonate and Sulfate*

Field observations revealed water seepage on the surface of some of the landforms, evidenced by the dark brown color of the sand layer compared to the surrounding dry, light greyish-yellow sand

layer (Figure 2a–f). In addition, a white film or sheet of precipitated salts was often observed on the leading edge and surface of the landforms (Figure 2c,d,h,i). The precipitated salts were in the form of flakes a few cm to tens of cm long and 1–2-mm thick, often containing bounded sand material. Due to repeated water seepage, the deposits were often stratified. The salt aggregations exhibited an arcuate distribution, a pronounced protrusion at the front and contraction at both sides, before gradually disappearing. They were common in low-lying areas but were not observed on the dunes, in agreement with previous research. As discussed later, their occurrence is the key to understanding the origin of these landforms.

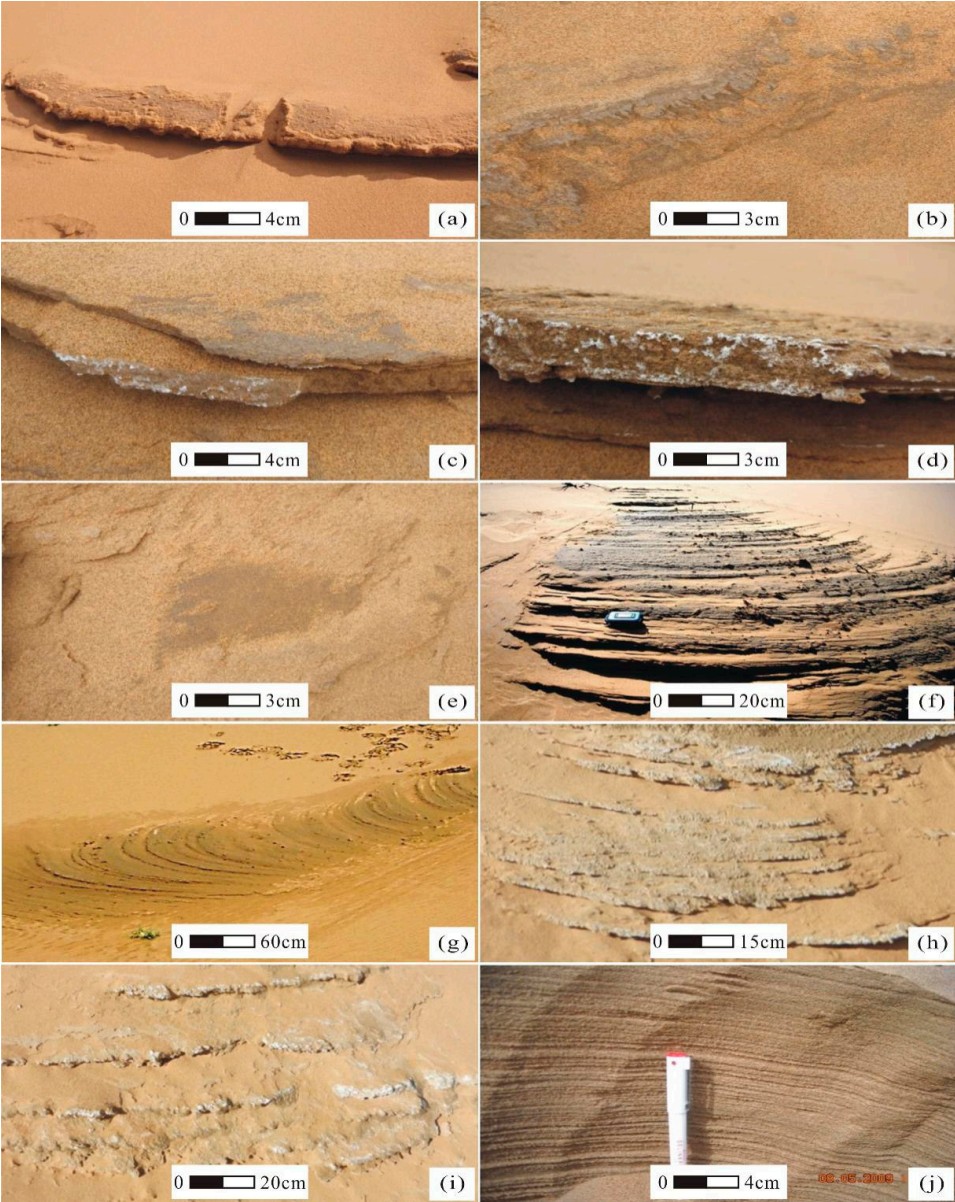

**Figure 2.** Microtopography on the slopes of the Badain Jaran megadunes. (**a**,**b**). Curved step with a dark surface in the upper part of the windward slope of the megadune to the east of Zhalate Lake. (**c**,**d**). Curved step with white precipitated $CaCO_3$ in the central part of the leeward slope of the megadune to the west of Yinderritu Lake. (**e**). Erosional runoff depression in the upper part of the windward slope of the megadune in the east of Zhalate Lake; note the dark base. (**f**). Multi-stage fan in the central part of the windward slope of the megadune to the east of Nuoertu Lake; note the dark surface. (**g**). Multi-stage fan in the upper part of the windward slope of the megadune to the east of Zhalate Lake; note the moist surface. (**h**,**i**). Multi-stage fan in the central part of the leeward slope of the

megadune to the west of Yinderritu Lake; note the white precipitated $CaCO_3$ and $CaSO_4$. (**j**). Sedimentary layers in a multi-stage fan in the upper part of the windward slope of the megadune to the east of Nuoertu Lake.

Electron microscopy observations revealed that the precipitated salts have a crystalline morphology, and that the crystal aggregates had a regular arrangement. There were three principal morphologies: columnar (Figure 3a), needle-like (Figure 3b), and fine crystalline (Figure 3c,d). Examination of the crystal morphologies indicated that the precipitated salts were rock salt, gypsum, or calcite. EDX analysis was conducted in on order to reliably determine their rmineralogical and chemical composition. The results revealed that they are mainly composed of $SO_3$ and CaO (Figure 4a–d, Table 1), with $Na_2O$, MgO, $Cl_2O$, and $SiO_2$ (Table 1) in some samples, and mainly $CO_2$ and CaO (Figure 4e,f, Table 1) in others. The high content of $SO_4$, $CO_2$, and CaO indicates that gypsum and calcite are the main mineral components. Some samples contained 5–11% $Na_2O$, and others also contained Cl. However, as the $SiO_2$ content was less than 4.8%, some of the Na was likely from NaCl; in other words, sodium was present.

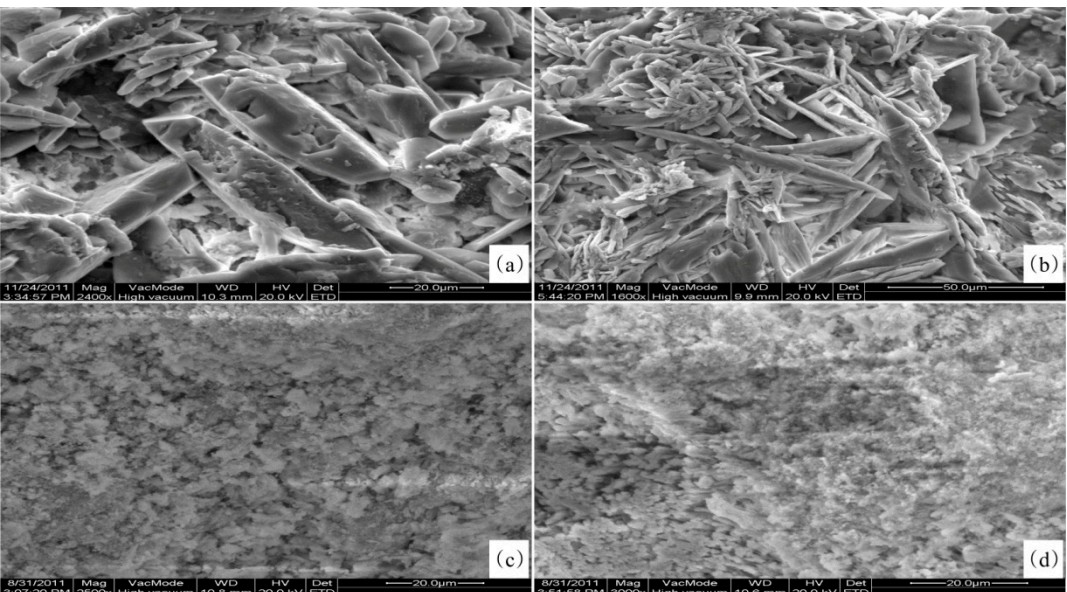

**Figure 3.** Crystalline morphology of precipitated salt in the multi-stage fans. (**a**). Columnar gypsum crystals in a multi-stage fan in the upper part of the windward slope of the megadune to the east of Zhalate Lake. (**b**). Needle-like gypsum crystals from the central part of the leeward slope of the megadune to the west of Yinderritu Lake. (**c,d**). Calcite micro-crystals of in a multi-stage fan in the upper part of the windward slopes of the megadunes to the east of Zhalate Lake and to the west of Yinderitu Lake.

**Table 1.** Chemical composition of precipitated salt samples obtained by energy spectrum analysis (%).

| Sites | Sample Number | CaO | $SO_4$ | $Na_2O$ | MgO | $SiO_2$ | $CO_2$ | $Cl_2O$ | $Al_2O_3$ |
|---|---|---|---|---|---|---|---|---|---|
| Zhalate | B01-004a | 36.95 | 51.66 | 5.27 | 0.94 | 2.61 | 0.51 | 2.06 | 0.00 |
| Zhalate | B01-004b | 32.45 | 52.97 | 6.12 | 2.34 | 4.67 | 0.50 | 0.95 | 0.00 |
| Huhejilin | B02-003a | 30.79 | 46.74 | 10.14 | 1.62 | 4.77 | 0.45 | 4.74 | 0.76 |
| Huhejilin | B02-003b | 34.38 | 52.33 | 6.19 | 1.33 | 4.50 | 0.41 | 0.86 | 0.00 |
| Yinderitu | BD-001 | 51.01 | - | - | 1.83 | 2.47 | 43.92 | - | 0.84 |
| Yinderitu | BD-002 | 44.13 | - | - | 2.48 | 2.67 | 49.15 | - | 1.21 |
| Zhalate | BD-003 | 35.87 | - | - | 1.58 | 1.16 | 60.90 | - | 0.49 |
| Zhalate | BD-004 | 35.71 | - | - | 1.54 | 1.62 | 60.23 | - | 0.43 |

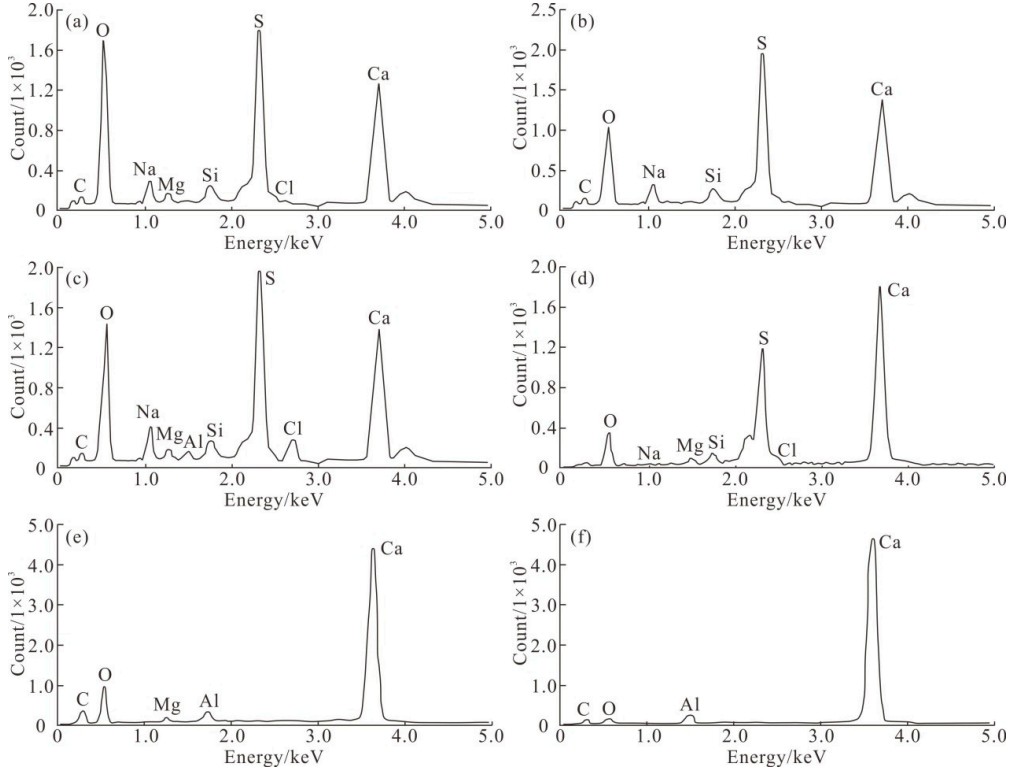

**Figure 4.** EDX results for precipitated salt samples from the multi-stage fans. (**a**,**b**), Megadune to the east of Zhalate Lake. (**c**,**d**), Central part of the windward slope of the megadune to the east of Huhejilin Lake. (**e**,**f**). Multi-stage fans to the east of Zhalate Lake and to the west of Yinderitu Lake.

## 3.5. Groundwater Overflow Zone at the Leading and Trailing Edges of Lakes

An investigation of Zhalate Lake, Badain Jaran Lake and other lakes revealed that a groundwater overflow zone (Figure 5a,b) is typically present at the lake edge. Water from the groundwater overflow zone discharges from the side close to the megadune and commonly forms a spring that enters the lake as a small stream (Figure 5a,b). The spring of the overflow zone has the characteristics of a gravity spring rather than rising water. The groundwater overflow zone mainly occurs on the leading and trailing edges close to the megadune; a groundwater overflow zone is absent at the edge of low dunes. The groundwater overflow zone is usually 1–3 m wide with a gradient of 2°–5°. The width of the groundwater overflow zone to the west of Zhalate Lake is ~1.8 m, and that of the groundwater overflow zone to the west of Badan Jaran Lake is ~2.3 m. Zones of herbaceous vegetation are present above the groundwater overflow zone.

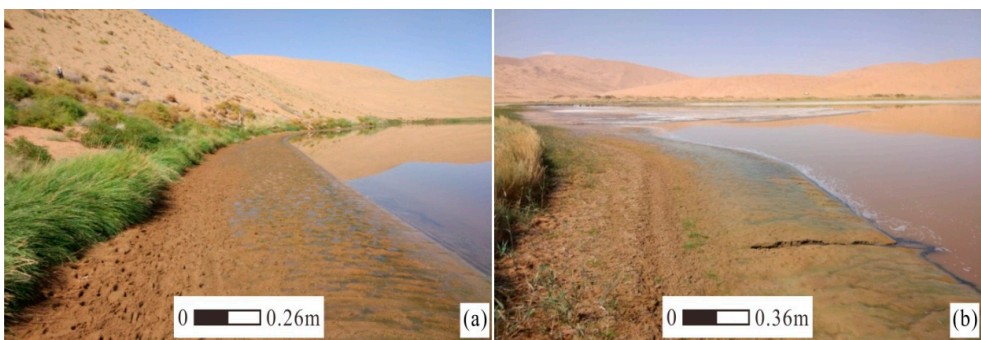

**Figure 5.** Groundwater overflow zone between megadune and lake. (**a**), Zhalate Lake; (**b**), western Badan Lake.

*3.6. Water Content and Seepage*

Field observations revealed the occurrence of water seepage on the surface of some of the studied landforms, evidenced by the dark-brown color of the sand layer compared to the surrounding dry, light greyish-yellow sand (Figure 2b,g). Water content measurements of 27 samples from each section (Figure 6) showed that the water content ranged from 0.2% to 5.0% (average of 2.6%) in section **a**; from 0.2% to 4.8% (average of 2.8%) in section **b** (Figure 6); from 0.1% to 4.2% (average of 1.9%) in section c (Figure 6); and from 0.01% to 4.4% (average of 2.6%) in section d (Figure 6). In the upper part of the megadunes to the east of Zhalate Lake, the maximum water content of the sand layer within two drill holes at 0.5-m depth was 7.8% and 7.6%, with respective averages of 6.7% and 6.4% (Table 2). The water content of the sand layer is usually very low (generally <3%) [24,29,30]; however, in the four studied sections, the water content in several sand layers was higher (3–5%). These findings show that precipitation can infiltrate to depths of 4 m or more in one year. The moisture within the sand layer at the depth of 0.4 m or more had not evaporated [24,26], and therefore moisture infiltrating to a depth of 4 m would be able to reach the groundwater and thus become a source of lake water.

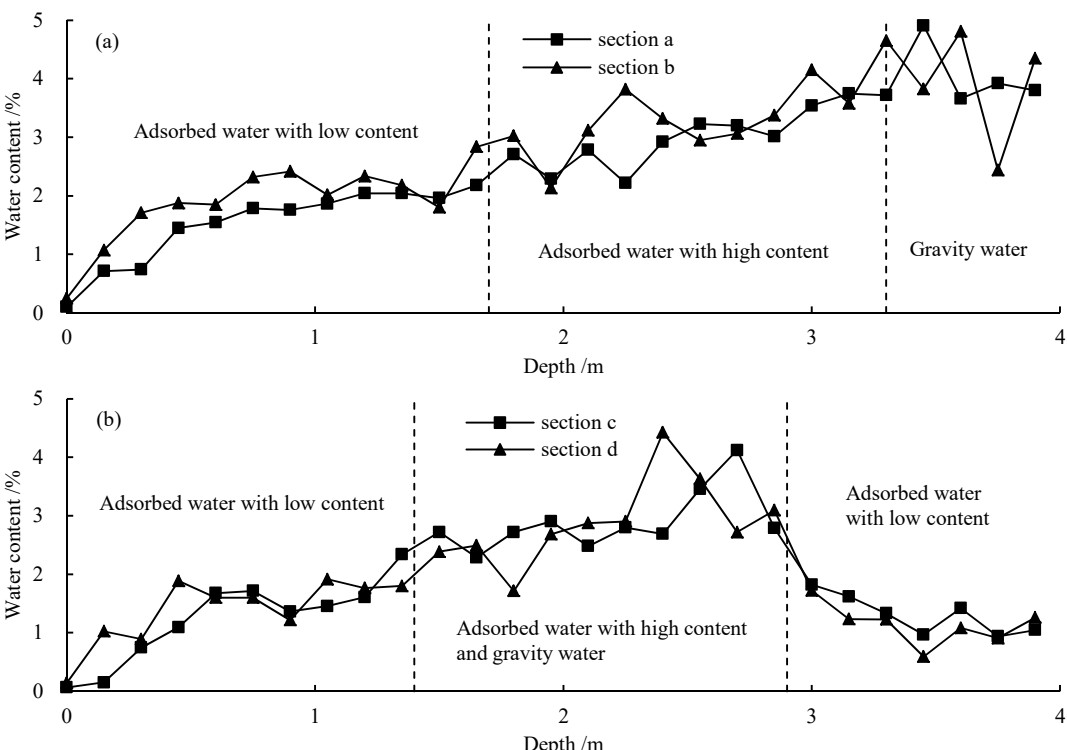

**Figure 6.** Depth profiles of water content in a sand layer in the leeward slope in the central part of the megadune to the west of Huhejaran Lake. (Water content more than 3.5% in sand is gravity water and water content less than 3.5% in sand is absorbed water).

**Table 2.** Moisture content in multi-stage fans in the megadune of eastern Zhalate Lake.

| Depth/cm | 5 | 15 | 25 | 35 | 50 |
|---|---|---|---|---|---|
| Moisture content in drill A (%) | 7.8 | 7.2 | 5.4 | 7.1 | 4.7 |
| Moisture content in drill B (%) | 6.3 | 6.7 | 7.2 | 7.6 | 5.6 |

*3.7. Particle-size Composition of the Studied Landforms and Aeolian sand*

The results of particle-size analysis of the 20 samples (Figure 7, Table 3) indicate that the main component of the coarse-grained layers of the multi-stage fans deposited from spring water is fine sand, with the content ranging from 45.8% to 51.5% (average of 49.6%); medium sand, ranging from 37.7% to

43.1% (average of 39.6%); and coarse and very fine sand, ranging from 6.0% to 8.7% (average 7.1%) and from 2.3% to 4.9% (average 3.9%). The main component of the fine-grained layers of the multi-stage fans is fine sand, with the content ranging from 56.8%–66.6% (average of 60.8%), followed by medium sand, ranging from 19.2% to 33.8% (average of 26.9%); and finally coarse and very fine sand, ranging from 1.8% to 5.8% (average 4.5%) and from 6.3% to 12.2% (average of 8.1%), respectively.

**Table 3.** Mean grain size content and parment of multi-stage fans in the megadune of eastern Nuoertu Lake (%).

| Sample Types | Sample Number | Coarse Sand (2.0–0.5 mm) | Medium Sand (0.5–0.25 mm) | Fine Sand (0.25–0.1 mm) | Very Fine Sand (0.1–0.05 mm) | Mean Mz | Mean σ | Mean Kg |
|---|---|---|---|---|---|---|---|---|
| Coarse particle layers of multi-stage fans | 10 | 7.1 | 39.6 | 49.6 | 3.9 | 1.94 | 0.76 | 0.96 |
| Fine particle layers of the multi-stage fans | 10 | 4.5 | 26.9 | 60.8 | 8.1 | 2.23 | 0.49 | 0.95 |
| Normal aeolian sand layer | 26 | 3.1 | 41.0 | 54.0 | 1.9 | 2.08 | 0.53 | 0.96 |
| surface coarse layer of mega dune | 26 | 12.6 | 47.5 | 39.4 | 0.5 | 1.83 | 0.56 | 0.96 |

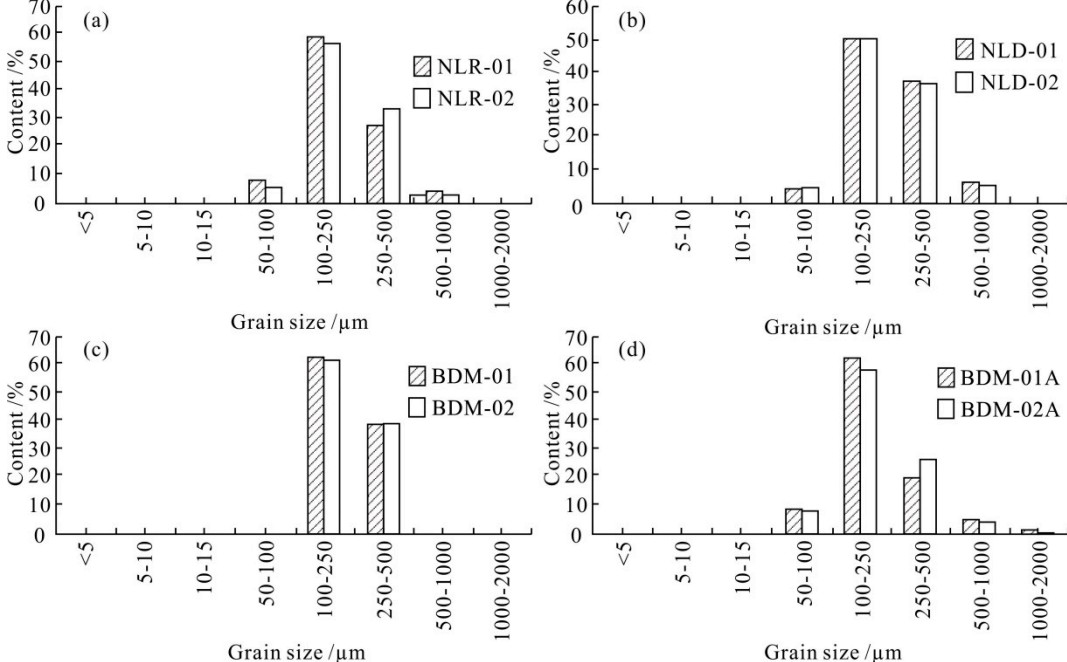

**Figure 7.** Particle-size distributions of samples from multi-stage fans. (**a**,**b**) Representative samples from fine-grained layers in a multi-stage fan (samples NLR01, NLR02) and from coarse-grained layers in a multi-stage fan (samples NLD01, NLD02) in eastern Nuoertu Lake. (**c**,**d**). Four representative samples of aeolian sand from a mega-dune in eastern Nuoertu Lake.

The results for the 26 aeolian sand samples collected from 10 cm below the surface of the megadunes showed that the composition was mainly fine sand, ranging from 30.6% to 73.9% (average of 54.0%); followed by medium sand, ranging from 20.0% to 66.2% (average of 41.0%); and finally coarse and very fine sand, ranging from 0 to 15.7% (average of 3.1%) and 0 to 9.7% (average of 1.9%), respectively (Table 3).

The results for the 52 aeolian sand samples from the megadune revealed a coarser particle-size composition of the surface residual coarse layer (reflecting the dispersion and loss of fine particles) compared to the underlying material (below 10 cm) (Table 3).

The coarse sand content of the coarse particle layer of the multi-stage fans was 4% higher than that of typical aeolian sand (Table 3), which indicates that the strength of hydrodynamic forces responsible for its formation exceeded that of the aeolian forces.

However, the fine particle layer of the multi-stage fans was finer-grained than that of the aeolian sand, indicating that the hydrodynamics forces (associated with springs) were weaker than the aeolian forces. All of the samples from the coarse and fine particle layers were close to a normal distribution and with unimodal or bi-modal particle-size characteristics. The particle-size frequency distribution curves and cumulative frequency distribution curves are characteristic of aeolian sand (Figure 8), which explains why multi-stage fans have developed in the aeolian sand. The particle-size distributions of the coarse particle layers of the multi-stage fans and the coarse surface layers of the megadunes are both bimodal (Figure 7; Figure 8) due to the increasing content of medium sand in the coarse particle layers, which are otherwise mainly composed of fine sand. The coarse particle layers and fine-grained layers alternate, indicating changes in spring flow, likely due to changes in precipitation. Particle-size parameters for all the sand types of the megadunes are listed in Table 3, and it can be seen that the average grain diameter of the sand layers ranges from 1.83 to 2.23Φ.

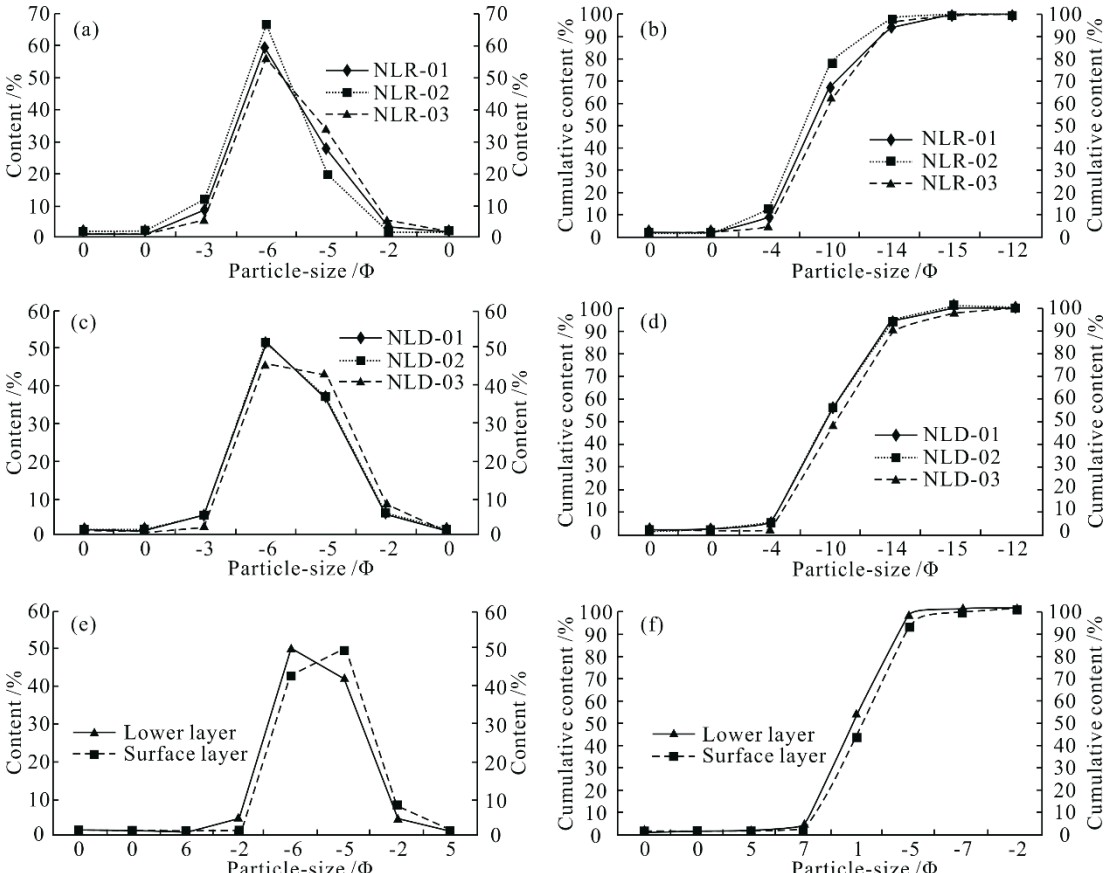

**Figure 8.** Particle-size frequency distribution and cumulative frequency distribution curves of samples from multi-stage fans and aeolian sand. (**a**,**b**) Fine-grained samples from a multi-stage fan in eastern Nuoertu Lake. (**c**), Coarse-grained samples from a multi-stage fan in eastern Nuoertu Lake. (**e**,**f**) Aeolian sand samples from eastern Nuoertu Lake. (Φ is defined as = −log2d. d is particle diameter (mm)).

The particle size of the fine sand layers in the multi-stage fans is finest and that of the coarse surface layer of the megadunes is the coarsest. The standard deviation of the coarse particle layers of the multi-stage fans is larger, which indicates that the degree of sorting of the coarse sand layer, formed by more intense spring activity, was less than that of the aeolian sand.

In addition, the standard deviation of the fine particle layers of the multi-stage fans is smaller than that of the aeolian sand, indicating that it is better sorted. Table 3 also shows that the material composition of multi-stage fans comes from aeolian sediments, and the grain size of surface layer

of aeolian sand deposits is coarser than that of lower layer, which is obviously different from the multi-stage fan which presents multi-layer changes of coarse and fine particle.

*3.8. Water Balance Assessment of the Megadune-Lake Region*

Water balance calculations can provide a basis for understanding the water sources of the study area. The soil water balance reflects the soil moisture content in an area. The annual soil water storage (W), annual precipitation (P), total evaporation (comprising both soil water evaporation and plant transpiration (E)), the amount of water intercepted by plants (I) and the amount of surface runoff (R) can be used to calculate the soil water balance. Annual soil water storage is equal to the annual rainfall minus losses from transpiration, surface runoff and total evaporation [25]. If W is positive, the soil water balance is positive and the input is greater than the losses, and vice versa if W is negative.

For the study area, Ma et al. [23] calculated the annual lake water recharge by precipitation. The new estimates provided herein are an improvement of that of Ma et al. [23] in three respects. First, the dune area used in the calculation of Ma et al. [23] was the area of bare sand, and the area of slopes with a gradient of >20° were included. On slopes with gradient >20°, a dry sand layer develops which may be 0.5 m or more in thickness, through which there will be very little precipitation infiltration. Thus, in order to obtain a reliable estimate, the area of slopes with a gradient >20° should be excluded from the calculation. A statistical analysis of the slopes in the megadune-lake region indicates that they represent 16.14% of the total area. Second, the lake area used in the calculation of Ma et al. [23] is 23 km$^2$; however, the area we used here is the average lake area throughout the year [31], which is 17.72 km$^2$. Third, the vegetation coverage used in Ma et al. [23] is 2%, which appears to be too low. Zhu et al. [32] investigated the megadunes within the desert region concludes that the vegetation coverage of megadunes does not exceed 5%, and therefore we used the value of 5%.

In order to estimate the groundwater recharge from precipitation in the megadune-lake region of the Badain Jaran Desert, the following data are required:

1. Total area of the megadune-lake region of the Badain Jaran Desert, which is ~29,242.7 km$^2$ [32].
2. Annual precipitation = ~90 mm.
3. Area of slopes with gradient >20° = 4719.77 km$^2$ (29,242.7 km$^2$ × 16.14%).
4. Average lake area throughout the year = 17.72 km$^2$ [31].
5. Area covered by megadunes vegetation = 1225.26 km$^2$ ((29,242.7 km$^2$ − 17.72 km$^2$ − 4719.77 km$^2$) × 5%).
6. Area of seepage springs = ~2.9 km$^2$ [23].
7. Area of bare sand dunes that can be used to calculate recharge. This was calculated as follows: 29,242.7 km$^2$ − 4719.77 km$^2$ − 1225.26 km$^2$ − 2.9 km$^2$ − 17.72 km$^2$ = 23,277.05 km$^2$.
8. Annual precipitation recharge of sand layer water. The annual rainfall in the region was average in 2010 (105 mm) and below average in 2011 (75 mm). During both 2010 and 2011 there were only two precipitation events exceeding 15 mm: 27.8 mm and 25.0 mm in 2010 and 19.7 mm and 15.8 mm in 2011 [33]. Based on the results of previous studies [34,35], such events are critical for determining whether or not the sand layer receives an effective water supply from precipitation. The annual recharge of precipitation to sand layer water for the period of 2010–2011 can be calculated as follows:

   [(25 mm − 15 mm) + (27.8 mm − 15 mm) + (15.8 mm − 15 mm) + (19.7 mm − 15 mm)]/2 = 14.2 mm.

   Because of the low precipitation in 2011, the annual recharge of precipitation to sand layer water in the region should be >14.2 mm, which represents only 15.8% available to recharge the groundwater. Notably, the recharge rate of soil water to groundwater is substantially lower than that in Mu Us sandy land [36].

9. Total annual evaporation capacity of seepage springs. Water surface evaporation for the megadune-lake region has been estimated as 1200–1550 mm [14,37]. The seepage springs can

maintain the megadunes microtopography in a wet condition for a long time; however, their annual evaporation will not exceed that from the water surface. Therefore, 1550 mm was used as the annual evaporation of seepage springs in the region. The total evaporation capacity of seepage springs is ~4,495,000 m$^3$ (2.9 km$^2$ × 1550 mm).

10. Total annual evaporation capacity of lakes. The total lake annual evaporation capacity is estimated to be 1550 mm [37]. Using this value, the total annual evaporation capacity of the lakes can be calculated as: 17.72 km$^2$ × (1550 mm − 90 mm) = 25,871,200 m$^3$.

11. Annual transpiration and evaporation of the vegetated area. In order to obtain reliable results, the mean annual rainfall value of 90 mm (for 2010 and 2011) was assumed to be equal to annual transpiration and evaporation in the vegetated area. Thus, total annual transpiration and evaporation of the vegetated area is 110,273,400 m$^3$ (1225.26 km$^2$ × 90 mm).

12. Total annual precipitation recharge of bare sand dunes, which can be used to calculate recharge. This is calculated as: 23,277.05 km$^2$ × 14.2 mm = 330,534,110 m$^3$.

From the foregoing, the amount of the remaining water can be calculated, as follows:

Amount of remaining water = 330,534,110 m$^3$ (total annual precipitation recharge of bare sand dunes) − 25,871,200 m$^3$ (total annual evaporation capacity of lakes) − 110,273,400 m$^3$ (annual evaporation and transpiration of the vegetated area) − 4,495,000 m$^3$ (total annual evaporation capacity of seepage springs) = 189,894,510 m$^3$.

It can be seen from the calculation above that the megadune-lake region with an area of 29,242.7 km$^2$ can maintain a rate of ~6493.7 m$^3$/km$^2$/year$^{-1}$ to recharge groundwater, and that atmosphere precipitation can effectively recharge groundwater after losses from evaporation, transpiration, interception by vegetation, and surface runoff. These results indicate a positive water balance in the megadune-lake region, and this result is quite different to that of Ma et al. [23].

## 4. Discussion

### 4.1. Landform Development and Water Source of the Megadunes

Research on landform development in the study region not only has a geomorphological significance but is also important for studying precipitation processes and effective precipitation. The erosional depressions in the area are caused by surface runoff. The infiltration rate of the sand layer is very high [34,38], some 10–20 times that of soil [30]. Thus, there is very little surface runoff in the region. A single rainfall event of more than 15 mm is classed as effective rainfall in desert areas [35]. Effective rainfall means that after consumption by evaporation and transpiration, there is sufficient remaining for deep penetration. When the precipitation of a single event is less than 15mm, it is generally lost completely via evapotranspiration [35]. Therefore, the development of erosional depressions with dark deposits at the bottom (Figure 2c) indicates the occurrence of individual rainfall events exceeding 15 mm and that surface infiltration-excess runoff is occurring. These heavy rainfall events have an additional function with regard to sand layer water and groundwater because of the high infiltration rate of the sand layer due its shallow depth (<0.4 m) [24].

Recent observations have indicated the occurrence of relatively heavy rainfall events in the area. In 2010, the total rainfall was average and rainfall events exceeding 15 mm occurred on two occasions. Precipitation reached 27.8 mm during one event in May, and 25 mm during an event in September [33]. The occurrence of effective rainfall means that is possible for rainfall to supplement the groundwater and thus the lake water.

Arcuate steps and multi-stage fans are formed by the occurrence of deep sand layer water concentrated within a specific depth range, from which water flows or seeps out during a major rainfall event. Aeolian sand will be moved forward when the deep sand layer water slowly flows out or seeps out, forming arcuate steps (Figure 2a–d) and multi-stage fans (Figure 2f–i).

Water seepage or wet sandy surfaces (Figure 2b,g,h) can be found on the surface of the arcuate steps or multi-stage fans, demonstrating that the water originates from slow flow or seepage. Previous

research has indicated that the content of $CaCO_3$, $CaSO_4$, and NaCl in sand deposits on the slopes of the megadunes is very low (<0.1%) [39].

The deposition and concentration of white salt deposits on the surfaces of arcuate steps and multi-stage fans deposits (Figure 2c,d,h,i),with a content sometimes exceeding 90% (Table 1), shows that these two landforms are formed by the slow flow or seepage of water.

The quantity of water flowing out from the deep layer controls whether an arcuate step or a multi-stage fan is formed: an arcuate step forms if the quantity of water flowing out is small, while a multi-stage fan forms if it is large. The development of arcuate steps and multi-stage fans therefore indicates that rainfall in the area penetrates to a substantial depth via seepage, that the sand layer water in the area has a positive balance, and that is quite possible for water to supplement the groundwater.

*4.2. Implications of the Water Content of the sand Layer*

The water content of the sand layer in desert areas with annual average rainfall <100 mm is normally <3% [24,29,34]. The water content of the sand layers in the study area is 3–5%, and in some sections the thickness of the sand layer with a moisture content of 3–5% can reach ~2 m (Figure 6). A moisture content of 3–5% thus has important implications for the existence of sand layer water which supplements the groundwater.

Previous research has shown that adsorbed water almost always supplements the groundwater and is often the main source [26,40]. The movement of adsorbed water is controlled by soil suction and it moves from the thick part of the water film to the thin part. Although usually slow, there may be large differences in the speed of water movement. In order to analyze the effectiveness of adsorbed water in supplementing the groundwater, it was categorized into adsorbed water with a high content (2.5–3.5%) and with a low content (<2.5%) (Figure 6).The film of adsorbed water with a high content has a large thickness and the water moves rapidly and is a major source of groundwater; the annual movement distance of high content adsorbed water can exceed 2 m in Chinese loess [30,40]. The direction of movement of adsorbed water is always downwards, because it is derived from rainfall. Low content adsorbed has a low film thickness and it moves slowly; it is either a minor supplement of groundwater and sometimes it may not supplement it at all. Gravity water is influenced by gravity and moves rapidly downwards within a soil; thus, it is the most important supplement of groundwater.

In the sand layer of the study area, gravity water and adsorbed water are present as a thick layer with a high water content, and they are an effective supplement of the groundwater and lake water. Each year, the seepage depth of rainfall in the area can penetrate at least 4 m according to the depth distribution of the gravity water (Figure 6) and the high adsorbed water content of the sand layer. Since the depth of the sand layer affected by evaporation is <0.4 m [24], water that seeps below this depth can move to the deep sand layer where it supplements the groundwater.

*4.3. Significance of the Groundwater Overflow Zone*

Although much research has been conducted on the water source of the Badain Jaran Desert, no information has been provided about the groundwater overflow zone [13,19–21]. It was argued in the foregoing that rainfall can supplement the groundwater in the study area; however, direct evidence of the supplementation of groundwater by rainfall would provide conclusive evidence for this. As mentioned earlier, there is typically a groundwater overflow zone on the leading and trailing edges of lakes, and running water of the overflow zone directly supplements the lake water or groundwater. The groundwater overflow zone in the area appears in the form of a gravity spring but without the characteristics of confined water which is moving upwards. It has been shown that groundwater in the area is not supplemented by a deep underground pathway [13].

Notably, the groundwater overflow zone on the leading and trailing edges of lakes occurs on one side, close to the megadunes (Figure 5), and there is no groundwater overflow zone on the side close to the low dune. Therefore, it can be determined that the water of the groundwater overflow zone in the area is derived from the megadune itself, namely from rainfall. This also indicates that groundwater

and lake water in the area is derived from rainfall. The sand layer on one side close to the low dunes is very thin and rainfall rapidly supplements groundwater after reaching the surface of the sand layer.

Therefore, there is no water in the low dunes, and it has been determined that there is no groundwater overflow zone on one side close to the low dunes. The importance of groundwater supplementation by the megadunes is also shown by the frequent occurrence of rills in the groundwater overflow zone.

### 4.4. Mechanism by Which Groundwater is Supplemented by Rainfall

Previous research has shown that the groundwater in the Chinese Loess Plateaus is supplemented by the effects of gravity and water film pressure [15,30,40]. Overall, the water in the megadune area in the Badain Jaran desert is in positive balance and rainfall is the source for supplementing groundwater and lake water. Five conditions are required for this: (i) rainfall in the area is concentrated in time and effective rainfall occurs during heavy rainfall events, which provides a water source that supplements the groundwater. (ii) A high content of gravity water and adsorbed water is produced by seepage into the sand layer, and the thickness of these two forms of water is large so that are able to supplement the groundwater. The gravitational action of gravity water and the pressure difference within the film of high content adsorbed water provide the forces necessary for the supplementation of the groundwater; moreover, these two forms of water are able to move rapidly. (iii) The infiltration rate of the sand layer is very high, which reduces rainwater evaporation and facilitates the migration of rainfall to deep water within the sand layer. (iv) The evaporation depth of the sand layer is shallow, which reduces evaporation and favors the conversion of sand layer water to groundwater. (v) The water binding capacity of the sand layer is weak which enables water to penetrate to the deep sand layer. A summary of the mechanism by which groundwater is supplemented by rainfall in the area is illustrated in Figure 9.

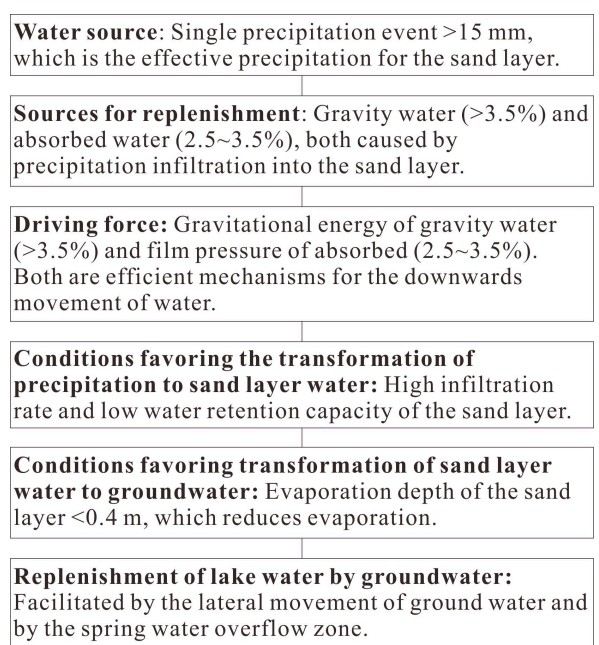

**Figure 9.** Summary of the mechanism of groundwater recharge by precipitation.

## 5. Conclusions

(1) Four distinctive landforms are developed in the megadunes area of the Badain Jaran Desert. Arcuate steps and multi-stage fans are formed by spring water activity and indicate that precipitation is the recharge source of water in the megadunes. Eroded depressions indicate that excess runoff is produced by heavy precipitation and that effective precipitation is provided by an individual

precipitation event exceeding 15 mm of rainfall. The groundwater overflow zone along lake shores indicates that precipitation is the recharge source of groundwater and that it supplies the groundwater and lake water by infiltration.

(2) As in other extremely arid desert regions, water leakage and the deposition of gypsum and calcite indicate the existence of a normal hydrological cycle and a positive water balance.

This means that not all of the surface water is lost to evapotranspiration, and that excess water infiltrates into the deep sand layer of the megadunes and becomes a water source for this layer.

(3) Gravity water and a high content of adsorbed water occur in the sand layer of the study area and it effectively recharges the groundwater and lake water. From the depth distribution of gravity water and the occurrence of a high content of adsorbed water at the depth of 4 m in the sand layer, it can be determined that the moisture within the sand is in a positive balance and the infiltration depth of precipitation can reach at least 4 m in one year.

(4) Water balance calculations reveal that, after accounting for evaporation, transpiration and other sources of loss of precipitation, a rate of ~6493.7 $m^3/km^2/year^{-1}$ remains to supply the groundwater and lake water.

(5) The recharge mechanism of groundwater by precipitation in the Badain Jaran Desert megadunes is directly linked to heavy precipitation events which provide a source of water, a high infiltration rate, shallow evaporation depth, and low water holding capacity; these factors act together to promote the transformation of precipitation to groundwater. The gravitational action of gravity water and the water film pressure resulting from a high content of adsorbed water provide the force necessary for effective water recharge.

**Author Contributions:** Conceptualization, D.-P.Y. and J.-B.Z.; funding acquisition, D.-P.Y.; investigation, Y.-D.M., X.-G.H., T.-J.S., D.-P.Y. and X.-Q.L.; methodology, T.-J.S. and J.-B.Z.; resources, T.-J.S. and X.-G.H.; software, X.-G.H. and A.-H.M.; validation, D.-P.Y., J.-B.Z., Y.-D.M. and T.-J.S.; writing—original draft preparation, D.-P.Y. and J.-B.Z.; writing—review and editing, D.-P.Y., J.-B.Z., and X.-Q.L.

**Funding:** This research was funded by the National Natural Science Foundation of China (41772180) and the Project of the State Key Laboratory of Loess and Quaternary Geology, Institute of Earth Environment, Chinese Academy of Sciences (SKLLQG1847, SKLLQG 1713).

**Acknowledgments:** The authors would like to thank Zhongdi Zhang at Shaanxi Normal University for assisting in the sample analysis, and thank all members who participated in the field survey in the Badain Jaran Desert.

**Conflicts of Interest:** The authors declare no conflict of interest.

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
