# Peer review of "Relationship between Landform Development and Lake Water Recharge in the Badain Jaran Desert, China"

_water, doi:10.3390/w11101999_

Round 1

Reviewer 1 Report

The authors describe in their publication entitled “Relationship Between Landform Development and Lake Water Recharge in the Badain Jaran Desert, China” the thorough investigation of a Chinese desert and its characteristics. The manuscript is written in a logical way but cannot be considered for direct publication due to a minor revision is needed:

Results:
3.4. SEM/EDX: The first paragraph of page 5 (lines 171-177) must be rewritten in terms of correct chemical compositions (for example SO42-, not SO3). The authors must also include respective (powder) XRD measurements to further characterize the different compounds and structures present in the sand. Furthermore, chemical compositions derived from EDX measurements should be avoided due to the lack of determining the bulk material. The authors should therefore reconsider Table 1 (line 200) and do bulk elemental analysis instead either by AAS or ICP-OES measurements.

Reviewer 2 Report

Review Report on

Manuscript ID: water-593680

General Conclusion:

The paper is recommended as acceptable after minor revision

Specific Comments

Throughout the entire paper it is recommended to use precipitation or Rainfall without "atmospheric": the additional "atmospheric" is needless, it is obvious.

The reference list is too long and could be shorter where just best citation is available. I leave it to the authors but example is enclosed: to delete refs no. : 2,3,5,6,9,10,15, 17, 19,20

It is important to distinguish between "Desertification" and "dryness".

These two terms are not identical: dryness is water scarcity and desertification of decline of soil fertility or productivity.

Change tallest to highest

Change fault to pathways (fault is a term used in geological tectonics)

Line 52; change modern to recent

Line 59: change complex to complexity

Line 60: change …is little… to …was poorly…

Line 68: change in Ma et al to :by Ma et al

Line 92: insert: by the climate of high barometric…

Line 94: change illumination to radiation

Figure 1 and within lines 111-120: add particle size range to coarse and fine

Line 120: add size in micron to particle size

Line 143: change concentrated to accumulated

Line 148: change leading to front

Line 156: insert: Secondary formation of Carbonate…

Line 161: change cemented to bounded

Line 170-171: Calcite and Gypsum can be easily chemically distinguished.

General comment to all Figures: to give the scale units in Cm (not in m)

Wide and thoroughly expanded of the outcome from the chemical analysis presented in Figure 4 are required , otherwise these information could be skipped.

Table 1: Insert units to its caption

There is a misunderstanding between two terms: underground overflow and runoff. Should be clarified.

Figure 6: A clarification between adsorbed and gravity waters might be essentially helpful.

Additional consideration of the case of capillarity and adsorption. The authors should state if upward capillarity of water within sand grains spaces is potentially possible.

Indication of the definition of Aeolian sand.

Table 3: Insert into the caption size of particles

Line 269: change inditing to indicating

Figure 8: Insert into the caption units and the definition of the symbol ɸ Lines 277-2823: expand the consideration how it is the outcome from figure 8

Water Balance issue: If balance is positive what is the fate of access waters

Line 356: what is the significance of "effective precipitation"?

Line 394-399: The fitted location of these lines is in "Material and Methods" and "Results"

The very detailed first paragraph about geomorphology, particle size distribution and more is not precisely inserted into the discussion.

Line 426: moving upward is negative gravity?

Throughout the entire paper: change water conversion to water migration (or movement or transition or transported) is recommended.

Figure9, The summary is excellent. It is recommended to allocate it in textual form as the abstract.
